# Mechanical Property Analysis and Calculation Method Modification of Steel-Reinforced High-Strength Concrete Columns

**DOI:** 10.3390/ma15196863

**Published:** 2022-10-02

**Authors:** Wenze Sun, Shiping Li

**Affiliations:** 1Department of New Materials and Building Engineering, Zaozhuang Vocational College, Zaozhuang 277000, China; 2Shanghai Key Laboratory for Digital Maintenance of Buildings and Infrastructure, Shanghai Jiao Tong University, Shanghai 200240, China

**Keywords:** high-strength concrete, steel-reinforced concrete column, bearing capacity, ductility, formula modification, finite element analysis

## Abstract

The existing studies lack research on the ductility of steel-reinforced high-strength concrete (SRHC) columns and current specifications restricted the use of high-strength concrete in steel-reinforced concrete (SRC) columns. To compensate for the shortcomings of the existing research and promote the application of high-strength concrete in SRC structures, we test six SRHC columns and one SRC column to examine the effects of the steel content, eccentric distance, and slenderness ratio on the ductility, bearing capacity, and failure mode of SRHC columns. Further, Abaqus finite element models are established to predict the influences of more parameters on post-peak ductility and analyze the relationship between strain development of the concrete and the decrease in bearing capacity of SRHC columns. The results show that the penetration of cracks into aggregate during failure is the primary reason for the poor ductility of the SRHC columns. Improving the confinement effect of the stirrups on concrete is the most effective measure to enhance the ductility of the SRHC columns. The decline in the stirrup spacing from 100 mm to 50 mm increased the ductility coefficient from 1.47 to 5.56. The effect of the steel content, stirrup strength, and slenderness ratio on the ductility coefficient of SRHC columns is less than 30%. After analyzing the reason for the error of current specifications, a modified formula with an error of less than 5% is developed.

## 1. Introduction

Steel-reinforced concrete (SRC) columns have been widely used in long-span spatial structures and ultrahigh-rise buildings [1,2,3]. Since the material properties of steel and concrete can be utilized simultaneously, these columns offer higher bearing capacity, ductility, seismic performance, and fire resistance than conventional reinforced concrete columns and pure steel columns [4,5,6]. However, the development of structures requires further enhancement of the bearing capacity and reduction of the self-weight of SRC members. High-strength concrete is an ideal building material because of its high strength, low creep, and excellent corrosion resistance [7,8,9,10,11]. Previous research has demonstrated replacing ordinary concrete with high-strength concrete in SRC columns can effectively reduce the building material usage and the section dimensions of members [12,13,14,15,16,17]. Moreover, the commercialization of high-strength concrete and the progress of pumping methods make it possible to apply high-strength concrete to SRC structures on a large scale.

Although steel-reinforced high-strength concrete (SRHC) columns have better mechanical properties and economic value than SRC columns, the high brittleness of high-strength concrete restricts its application in practical engineering. Current design specifications limit the maximum concrete strength of the SRC columns to prevent the brittle failure of structures caused by the increased concrete strength: EN1994-1-1-2004 [18] and AISC360-16 [19] stipulate that the maximum strength of concrete (*f*_c_) should not exceed 75 and 70 Mpa, respectively. JGJ138-2016 [20] requires that the maximum compressive strength of concrete should equal 80 MPa, and AIJ [21] stipulates that the maximum strength of concrete is 90 MPa. In order to promote the application of high-strength concrete in SRC structures, scholars have conducted extensive research on the mechanical property of SRHC columns and verify the applicability of the existing specifications.

In past few years, a considerable number of studies have been carried out on mechanical property of SRHC columns, including compressive capacity [22,23,24], flexural behavior [25] and seismic performance [26,27,28]. In recent years, scholars have focused on methods of calculating various specifications of SRHC columns. In 2011, Ellobody et al. [29] presented a nonlinear finite element model for studying the bearing capacity of slender SRHC columns under different eccentricities. They found that when the eccentricity (*e*) equaled 0.125 of the overall depth of the column sections (*D*), the steel strength significantly affected the bearing capacity of the SRHC columns; however, when *f*_c_ ≧ 70 MPa and the eccentricity was 0.375*D*, the effect of steel strength on the bearing capacity of the SRHC columns was no longer noticeable. Moreover, they found that EN1994-1-1-2004 could accurately predict the bearing capacity of both SRC and SRHC columns. In 2012 and 2014, Kim et al. [30,31] analyzed 14 SRHC columns with high-strength steel and compared the mechanical properties of the columns with different steel forms. They found that the bearing capacity and ductility of concrete-filled steel tube columns or concrete-encased L-shaped steel columns were remarkably higher than those of concrete-encased H-shaped steel columns. In addition, they found that the bearing capacity of the SRHC columns evaluated by code AISC360-16 [19] was precise compared to test results; in contrast, the bearing capacity assessed by the EN1994-1-1-2004 [18] provision was unsafe.

In 2018, Hung et al. [32] found the addition of steel fiber can effectively inhibit the cracking of concrete but the effect on ductility is not obvious with a 0.75% volume fraction. And they reported that ACI 318 [33] overestimates the bearing capacity of slender columns with high-strength concrete about 10 %. In 2019, Lai and Liew [34] studied the axial compression stability of slender SRHC columns using experimentation and finite element analysis. They reported that codes EN1994-1-1-2004 and AISC360-16 provided a conservative prediction of the bearing capacity and flexural stiffness of SRHC columns. The slenderness ratio and steel content markedly influenced the accuracy of the specifications in predicting the bearing capacity of the columns. However, in another study, Lai and Liew [35] stated that specification EN1994-1-1-2004 overestimated the bearing capacity of short SRHC columns and revealed that an insufficient constraint on concrete caused this situation. In 2022, Du et al. [36] reported that both codes EN1994-1-1-2004 and JGJ138-2016 [18,20] might produce unsafe strength predictions for SRHC columns under axial compression, but the predictions were conservative under eccentric compression. Research on post-peak ductility was conducted by scholars on the high-strength concrete-filled steel tube columns. In 2021, Phan et al. [37] conducted a dynamics simulation analysis of high-strength concrete-filled steel tube columns. The contribution of the steel and the depth-to-thickness ratio significantly impacted the ductility of the columns, and code EN1994-1-1-2004 could accurately predict the ultimate compressive strength of high-strength concrete-filled steel tube columns. Xiong et al. [38] found that both EN1994-1-1-2004 and AISC360-16 underestimated the compressive strength of high-strength concrete-filled steel tube columns and that the codes AIJ and CISC [39] may give an unsafe prediction of the axial strength for these columns; they reported that the addition of steel fibers may decrease the ductility of those columns. In the latest research, scholars have focused on sustainability of high-performance concrete structure. In 2020, Konecny, P. and Hrabova, K. [40] demonstrated the importance on sustainability of RC structures in an environment with chloride ion erosion and offered the best concrete mix proportion in the view of sustainability assessment. In 2021, K. Hrabova and B. Teply [41] proposed a method for quantifying and comparing the concrete mix variants for different characteristics, and the relationship of the service life, performance, and eco-costs has been defined for the HPC sustainability.

Even though a large number of studies have been conducted on SRHC columns, the previous research demonstrates that there are still deficiencies in two aspects:The existing research on the accuracy of the specifications mainly focuses on slender columns and lacks validation of their accuracy for middle columns. Furthermore, the different research conclusions about the accuracy of the codes predicting the bearing capacity are inconsistent.Previous studies have not sufficiently investigated the ductility of high-strength concrete-encased steel columns. Therefore, more research is required to examine the deformation capacity of SRHC columns and the accuracy of the code predictions.

The present study aims to compensate for the shortcomings of the existing research on the ductility of SRHC columns and propose revisions to improve the accuracy of the current specifications on predicting SRHC columns bearing capacity. This paper conducts compression tests on middle SRHC columns made of C100 concrete and Q355 steel and investigates the impacts of the concrete strength, steel content, eccentric distance, and slenderness ratio on the ductility and bearing capacity of SRHC columns. Furthermore, the failure mode is observed to study the mechanism for the brittle failure of the SRHC columns. Then, an Abaqus analytical model for the SRHC columns is developed to examine the effect of more factors on the ductility of the columns. In addition, this paper compares the results calculated by specifications EN1994-1-1-2004, JGJ138-2016, and AISC360-16 with the test data to verify their accuracy. Revisions to improve the accuracy of the specifications are finally proposed. The main research methods and processes are described in Figure 1.

## 2. Experimental Investigation

### 2.1. Testing Specimens

This work tests six steel-reinforced high-strength concrete (SRHC) columns and one steel-reinforced ordinary concrete (SRC) column. Table 1 tabulates the critical parameters, including the concrete strength, the steel content, the eccentric distance, and the slenderness ratio. The sections of all the columns have the dimensions 200 mm × 200 mm, and Figure 2 shows the dimensions and reinforcement form of the section steel.

### 2.2. Material Properties

This work uses C50 and C100 concrete grades, and Table 2 lists the proportion of C100 concrete. According to code GB50152-2012 [42], the average compressive strengths of C50 and C100 concrete grades cured under the same conditions for 28 days are 55 and 106 MPa, respectively, with standard deviations of 1.36 and 1.92, respectively.

This work adopts Q355 steel for the H-shaped steel and HRB400 for the reinforcement. According to code GB/T228-2010 [43], three samples of every type of H-shaped steel and reinforcement were selected for the material tests. Table 3 lists the mechanical properties of the steel and the reinforcement.

### 2.3. Testing Setup and Procedure

A 10,000 kN automatic testing machine (Dongce Testing Machine Technology Co., Ltd., Jinan, China) illustrated in Figure 3 conducted the compression tests. The axial load was directly applied to the column under axial compression through the top plate, while the eccentric load was applied to the specimen utilizing a knife hinged on the top of the column. Three layers of carbon-fiber cloth were wrapped around the 1/6 height of the top of the column to prevent its top from being prematurely damaged.

Figure 4 depicts the arrangement of the measuring points. Strain gauges (Jingming Technology Co., Ltd., Yangzhou, China) were placed halfway up the column to measure the strain of the concrete, stirrups, and steel sections. The measured results of strain will be used for the mechanism of the effect of different parameters on the mechanical properties of SRHC columns. As shown in Figure 3, the four vertical displacement gauges (Liyang Instrument Factory, Liyang, China) were arranged to record the vertical displacement of all specimens and horizontal displacement gauges were arranged to record the lateral deflection of the eccentric specimens.

The tests were carried out under graded loading according to code GB50152-2012 [39]. Before formal testing, 100 kN preloading was adopted to test the performance of the testing machine. In the formal tests, the load increment of each stage was 10% of the estimated peak load at a rate of 300 kN/min, and the load was held for 3 min in each stage. The loading rate was reduced to 150 kN/min after the load reached 60% of the estimated ultimate bearing capacity of the column. When the load reached 80% of the estimated ultimate bearing capacity of the column, the displacement control method at a rate of 0.6 mm/min was employed for loading. After the peak load was reached, the test was terminated when the load dropped below 70% of the peak load.

## 3. Test Results and Analysis

### 3.1. Failure Mode

Figure 5d shows the failure mode of ordinary-strength concrete column A4 (C50). The damage is located in the middle of the column, and a high proportion of concrete cracks are in this area. Moreover, the concrete in the crushed zone indicates significant fragmentation, but no large areas of spalling concrete are present. After the specimen fails, the concrete confined by the stirrups is not crushed. The failure mode of steel-reinforced high-strength concrete columns differs from that of the SRC column under axial compression. According to Figure 5a–c, the damage occurs at one end of the columns. The specimen is damaged with only a few large cracks on the concrete surface and accompanied by extensive spalling of the concrete. Removing the failed concrete reveals that the concrete confined by the stirrups is remarkably crushed.

Figure 6 depicts the failure mode of the columns with various eccentric distances. For specimens E1 with an eccentric distance of 40 mm, fine cracks form on the tensile side under 20% of the ultimate load and do not develop further as the load increases. The specimen fails with a loud bang, and there is no visible bending. When specimen E2 with an eccentric distance of 120 mm is damaged, it has obvious bending deformation. There are several horizontal cracks on the tensile side of the specimen, and the location of the cracks is consistent with that of the stirrups. Compared with specimen A3 under axial compression, the concrete of specimen E1 with an eccentric distance of 40 mm confined by the stirrups is less damaged, and that of specimen E2 with an eccentric distance of 120 mm is not damaged. Therefore, increasing the eccentric distance reduces both the degree of damage to concrete confined by the stirrups and the influence of concrete on the failure of the columns gradually.

From the form of damage to the SRHC columns, it can be concluded that the hoop reinforcement is much less effective in restraining high-strength concrete than ordinary concrete. This can cause the high-strength concrete inside the hoop to be too damaged to provide residual strength for the specimen at the time of the failure of the concrete protective layer, leading the column to be significantly brittle when failing. Moreover, as the damage to the concrete aggregate in Figure 7 demonstrates, due to the excessive strength of the cementitious material in high-strength concrete, the aggregate is crushed or cracked when the concrete is damaged, giving rise to lower energy consumption during damage. This can be another cause of the brittle failure of the SRHC columns.

### 3.2. Load–Displacement Curve and Stiffness

Figure 8a delineates the load–displacement (*N*–Δ) curves of the columns with various concrete strengths. Before the *N*–Δ curves reach the ultimate load, the stiffness of specimen A3 (C100) is much higher than that of specimen A4 (C50). Furthermore, when the *N*–Δ curves approach the ultimate load, stiffness remains unchanged for the SRHC columns but remarkably decreases for ordinary concrete ones, which is because the SRHC columns have high stiffness owing to the high density of high-strength concrete. Additionally, the yielding of the steel and reinforcement has a negligible effect on the stiffness of the SRHC columns during the loading process. Therefore, the stiffness of the SRHC columns is high and remains constant.

Figure 8b shows the load–displacement curves of the columns with different steel contents. The slopes of the *N*–Δ curves are similar under different steel contents. After the load–displacement curves reach the ultimate load, the bearing capacity of all the specimens decreases significantly. However, the bearing capacity of the columns with a high steel content declines less because raising the steel content can improve the constraining effect on concrete so that the constrained part can provide a higher bearing capacity.

As shown in Figure 8c, with an increase in the eccentric distance, the stiffness of the column decreases noticeably. Additionally, the stiffness of each SRHC column under eccentric compression remains constant before the *N*–Δ curves reach the ultimate load. Figure 8d shows the load–displacement curves of the columns with different slenderness ratios. Reducing the slenderness ratio can enhance stiffness but has no substantial effect on the descending segment of the *N*–Δ curves.

### 3.3. Bearing Capacity and Ductility

In order to explore the influence of different parameters on the ductility of SRHC columns, we introduce the ductility coefficient [44] to analyze the deformability of the columns, as expressed in Equation (1). Figure 9 plots the strain curves of the steel and concrete of the typical specimens to examine the mechanism for the influence of various parameters on the ductility of the columns. Table 4 presents the test results of the bearing capacity and ductility coefficient of the SRHC columns.
*μ* = Δ_f_/Δ_y_(1)
where Δ_f_ is the limit displacement, and Δ_y_ indicates the nominal yield displacement.

Figure 10 shows the specific meaning of the displacement ductility coefficient.

Comparing the test results of the specimens listed in Table 4 reveals that increasing the steel content can enhance the bearing capacity of the SRHC columns, albeit not remarkably. When the steel content rises from 3.63% to 5.13% and 6.20%, the ultimate bearing capacity of the columns increases by 8.7% and 9.6%, respectively. Changing the strength of the concrete influences the bearing capacity of the columns markedly. For instance, raising the concrete strength from C50 to C100 increases the ultimate bearing capacity of the columns by 30.7%. It is worth noting that due to the good constraining effect of the stirrups on ordinary concrete and the stress concentration phenomenon of the SRHC columns, the increasing degree of the bearing capacity of the specimens is much lower than that of concrete strength. The eccentric distance is the factor affecting the bearing capacity of the columns most apparently. Increasing the eccentric distance of the columns from 0 mm to 40 and 120 mm reduces their ultimate bearing capacity by 23.5% and 75.7%, respectively. The slenderness ratio does not markedly impact the bearing capacity of the columns. For example, the bearing capacity of the specimen with a height of 600 mm increases by less than 3% compared to that of specimen A3, which has a height of 1200 mm, which can be ignored.

It can be concluded from Table 4 that the influence of the concrete strength on the ductility of the columns is substantial. When the concrete strength decreases from 106 to 55 MPa, the ductility coefficient of the columns rises by 35.3%. The influence of the steel content on the ductility of the columns is limited. As the steel content increases from 3.63% to 5.13% and 6.20%, the ductility coefficient of the columns enlarges only by 14.9% and 21.5%, respectively. Figure 9a demonstrates that since the Q355 steel yields before the concrete is broken, the steel can only constrain the concrete and cannot provide residual strength for the column. Therefore, the steel ratio affects the ductility coefficient of the columns negligibly.

Furthermore, the ductility coefficient of the eccentrically stressed columns is related to the state of force on the sections. When the eccentricity of the SRHC columns rises from 0 to 40 mm, their ductility coefficient declines by 19%, but when it increases from 40 to 120 mm, their ductility coefficient enlarges by 28%. Figure 9b shows that the columns with an eccentricity of 40 mm are still subjected to compression failure. Similar to the columns under axial compression, the steel flange under compression yields before the concrete failure and cannot provide residual strength for the column. In addition, the confinement effect of the stirrups and steel on concrete is much lower under eccentric compression than under axial compression, resulting in a ductility coefficient lower than that of the columns under axial compression. When the eccentricity increases to 120 mm, the failure mode of the column changes to tensile failure, and the influence of concrete on the column declines. Figure 9c demonstrates that the steel flange under tension does not yield when the concrete damages and the steel still has residual tensile strength; thus, the ductility coefficient of the column increases significantly. Nevertheless, the slenderness ratio slightly affects the ductility coefficient of the columns. When the height of the columns declines from 1200 to 600 mm, their ductility coefficient varies by less than 2%.

## 4. Finite Element Parametric Analysis

Due to the high brittleness of high-strength concrete, the bearing capacity of all the steel-reinforced high-strength concrete columns declines suddenly upon failure. Furthermore, the ductility coefficient of the SRHC columns cannot be improved significantly by changing the steel content, the eccentric distance, and the slenderness ratio. Therefore, in order to change the brittle failure of SRHC columns, we should consider more factors that improve their ductility. According to Shi et al. [45], concrete confined by lower stirrup spacing offers good ductility, and the ductility of high-strength concrete can also be enhanced by increasing the stirrup strength. Kim et al. [30] also reported that SRHC columns containing high-strength steel had good ductility under eccentric compression. Therefore, an Abaqus model should be established to study the impacts of stirrup spacing, stirrup strength, and steel strength on the ductility of SRHC columns.

### 4.1. Model Establishment

The model uses the reduced integral solid element (C3D8R) for the concrete and the steel and employs the truss element (T3D2) for the reinforcement. According to Ellobody’s study [1], the coefficient of the bond between H-shaped steel and concrete is 0.25. The concrete is divided into three parts according to different constraints and is endowed with various material properties, as shown in Figure 11a. The boundary conditions of the specimens are the same as the actual situation.

The SRHC column model uses the broken line model for the steel and the reinforcement, and their material properties are measured according to the tests. Since the compressive performance of concrete significantly improves after being constrained, this model defines the performance of concrete materials according to the stress–strain curve obtained by Mander et al. [46]. The formula for the constrained concrete is defined as:(2)$$\sigma =\frac{f_{cc}xr}{r-1+x^{r}}$$
(3)$$f_{cc}=kf_{c0}$$
(4)$$x=\varepsilon /\varepsilon_{cc}$$
(5)$$r=E_{c}/(E_{c}-E_{\sec})$$
(6)$$\varepsilon_{cc}=\left[1+5(k-1)\right]\varepsilon_{c0}$$
where *σ* and ε are the stress and strain on the concrete, respectively; *E*_sec_ and *E*_c_ represent the secant modulus and elasticity modulus of the concrete, respectively; *f*_c0_ is the axial compressive strength of the unconstrained concrete; *k* is the improvement coefficient on constrained concrete strength.

### 4.2. Verifying Finite Element Model Rationality

Table 5 compares the simulated bearing capacity of typical specimens with the actual data. The maximum and average differences between the simulated and actual bearing capacity of the typical specimens are –2.92% and –0.93%, respectively, with a standard deviation of 1.56%. Hence, the bearing capacity simulation is relatively accurate. Figure 12 also compares the simulated curve with the test curve. The displacements corresponding to the simulation and test curves are roughly the same when the ultimate bearing capacity is reached. Further, the simulation and test curves basically have a similar form. Therefore, the finite element model considering the confinement effect of the concrete can simulate the stress on the SRHC columns well, and this model can be used for further parametric analysis.

### 4.3. Analysis of Finite Element Results

This paper utilizes the Abaqus model to study the effects of the stirrup spacing, stirrup strength, and steel strength on the ductility of SRHC columns. Table 6 lists the simulation results of the finite element expansion parameters.

Table 6 and Figure 13a demonstrate that reducing the stirrup spacing significantly improves the ductility coefficient of the SRHC columns. Reducing the stirrup spacing from 100 mm to 75 and 50 mm raises the ductility coefficient of the SRHC columns by 37% and 278%, respectively. When the stirrup spacing drops to 25 mm, the *N*–Δ curve does not decrease after concrete crushing, and only the growth rate of the bearing capacity decreases; further, Equation (1) cannot calculate its ductility factor. Figure 13c shows that the stirrups yield when the bearing capacity of the column decreases to 80% of the maximum bearing capacity, indicating that the smaller stirrup spacing can give full play to its material properties and provide a reasonable constraint on the high-strength concrete.

Table 7 and Figure 14a demonstrate that increasing the strength of the stirrups can enhance the ductility coefficient of the SRHC columns, albeit limitedly. Moreover, with an increase in the stirrup strength, the increase in the ductility coefficient of the SRHC columns shows a decreasing trend. Raising the stirrup strength from 477 MPa to 675, 900, and 1150 MPa enlarges the ductility coefficient of the SRHC columns by 19%, 36%, and 46%, respectively. According to Figure 14c, the stirrups do not reach the yield strength when the bearing capacity of specimen S7 decreases to 80% of the maximum bearing capacity. This implies that employing higher stirrup strength wastes materials, and the improvement in the confinement effect of concrete by increasing the stirrup strength is far less than that by reducing the stirrup spacing.

Figure 15a shows that the ductility coefficient of the SRHC columns correlates nonlinearly with their steel strength. When the steel strength increases slightly, the ductility coefficient of the columns does not improve significantly, but their bearing capacity can improve. As the steel strength increases, its influence on the ductility coefficient of the SRHC columns becomes more and more prominent. For example, when the strength of steel reaches 1000 MPa, the decrease in the bearing capacity of the SRHC columns after failure is less than 20%. According to Figure 15c, since the yield strain of high-strength steel (*ε*_a_ ≥ 3500 µε) is much higher than that of concrete (*ε*_cu_ ≈ 2800 µε) under failure, when the concrete is crushed, the high-strength steel does not yield. As a result, the core column composed of high-strength steel and concrete inside the steel can continue to bear the load, enhancing the ductility of the SRHC columns noticeably.

## 5. Calculations and Analysis of Different Specifications

### 5.1. Calculation Methods

The code “Specification for Structural Steel Buildings” (AISC360-16) [19] calculates the bearing capacity of steel-reinforced concrete columns according to pure steel structures by converting the reinforced concrete part to equivalent steel. The specification offers a conservative method to calculate the bearing capacity as follows:(7)$$\left\{ \begin{array}{ll} \frac{P_{r}}{P_{n}}+\frac{8M_{r}}{9M_{n}}\leq 1.0 & \frac{P_{r}}{P_{n}}\geq 0.2\\ \frac{P_{r}}{2P_{n}}+\frac{M_{r}}{M_{n}}\leq 1.0 & \frac{P_{r}}{P_{n}}\leq 0.2 \end{array}\right.$$
(8)$$\left\{ \begin{array}{ll} P_{n}=P_{n0}\left[0.658^{\left(\frac{P_{n0}}{P_{e}}\right)}\right] & \frac{P_{n0}}{P_{e}}\leq 2.25\\ P_{n}=0.877P_{e} & \frac{P_{n0}}{P_{e}}\geq 2.25 \end{array}\right.$$
(9)$$P_{n0}=f_{y}A_{s}+f_{\mathrm{ysr}}A_{\mathrm{sr}}+0.85f_{c}{}^{\prime }A_{c}$$
where *P*_r_ and *P*_n_ indicate the bearing capacity of the SRC column under eccentric and axial compression, respectively; *M*_r_ is the flexural bearing capacity; *M*_n_ denotes the flexural bearing capacity in a pure bending state; *P*_e_ is the elastic critical buckling load; *A*_c_, *A*_s_, and *A*_sr_ represent the sectional area of the concrete, steel, and reinforcement, respectively; and *f*_c_’, *f*_y_, and *f*_ysr_ indicate the compressive strength of the concrete, steel, and reinforcement, respectively.

The formulas for calculating the bearing capacity of the SRC columns under axial and eccentric compression provided by the code “Design of Composite Steel and Concrete Structures” (EN1994-1-1-2004) [18] in Europe are expressed in:(10)$$N_{\mathrm{Ed}}\leq \chi N_{\mathrm{pl},\mathrm{Rd}}$$
(11)$$\left\{ \begin{array}{ll} \frac{N_{u}-N_{\mathrm{pm},\mathrm{Rd}}}{N_{\mathrm{pl},\mathrm{Rd}}-N_{\mathrm{pm},\mathrm{Rd}}}+\frac{M_{u}/\alpha_{M}}{M_{\mathrm{pl},\mathrm{Rd}}}=1 & \;(AC)\\ \frac{N_{u}-0.5N_{\mathrm{pm},\mathrm{Rd}}}{0.5N_{\mathrm{pm},\mathrm{Rd}}}+\frac{M_{u}/\alpha_{M}-M_{\mathrm{pl},\mathrm{Rd}}}{M_{\max,\mathrm{Rd}}-M_{\mathrm{pl},\mathrm{Rd}}}=1 & \;(CD)\\ \frac{N_{u}}{0.5N_{\mathrm{pm},\mathrm{Rd}}}+\frac{M_{u}/\alpha_{M}-M_{\max,\mathrm{Rd}}}{M_{\mathrm{pl},\mathrm{Rd}}-M_{\max,\mathrm{Rd}}}=1 & \;(BD) \end{array}\right.$$
(12)$$\left\{ \begin{array}{l} N_{\mathrm{pl},\mathrm{Rd}}=0.85f_{c}{}^{\prime }A_{c}+f_{y}A_{a}+f_{s}A_{s}\\ N_{\mathrm{pm},\mathrm{Rd}}=0.85f_{c}{}^{\prime }A_{c}\\ M_{\max,\mathrm{Rd}}=0.85W_{c}f_{c}{}^{\prime }/2+W_{a}f_{y}+W_{s}f_{s}\\ M_{\mathrm{pl},\mathrm{Rd}}=M_{\max,\mathrm{Rd}}-0.85W_{\mathrm{hc}}f^{\prime }{}_{c}/2-W_{\mathrm{ha}}f_{y} \end{array}\right.$$
where *N*_ed_ and *N*_u_ are the bearing capacity of the SRC column under eccentric and axial compression, respectively; *M*_u_ is the flexural bearing capacity; α_M_ denotes the reduction coefficient of the flexural capacity; *f*_y_, *f*_s_, and *f*_c_’ indicate the compressive strength of the steel, reinforcement, and concrete, respectively; *W*_c_, *W*_a_, and *W*_s_ represent the bending section coefficient of the concrete, steel, and reinforcement, respectively; and *W*_hc_ and *W*_ha_ stand for the bending section coefficient of the concrete and steel in 2*h*_n_, respectively.

The “Code for Design of Composite Structures” (JGJ138-2016) [20] is based on the limit equilibrium method, and Figure 16 shows the stress distribution of the section. The calculation formulas are expressed by:(13)$$N\leq 0.9\varphi (f_{c}A_{c}+{f^{\prime }}_{y}{A^{\prime }}_{s}+{f^{\prime }}_{a}{A^{\prime }}_{a})$$
(14)$$\left\{ \begin{array}{l} N_{u}\leq \alpha_{1}f_{c}bx+f^{\prime }{}_{y}A^{\prime }{}_{s}+f^{\prime }{}_{a}A^{\prime }{}_{\mathrm{af}}-\sigma_{s}A_{s}-\sigma_{a}A_{\mathrm{af}}+N_{\mathrm{aw}}\\ N_{u}e\leq \alpha_{1}f_{c}bx(h_{0}-\frac{x}{2})+f^{\prime }{}_{y}A^{\prime }{}_{s}(h_{0}-a^{\prime }{}_{s})+f^{\prime }{}_{a}A^{\prime }{}_{\mathrm{af}}(h_{0}-a^{\prime }{}_{a})+M_{\mathrm{aw}} \end{array}\right.$$
(15)$$e=e_{0}+e_{a}+\frac{h}{2}-a$$
where *N* and *N*_u_ represent the bearing capacity of the composite column under eccentric and axial compression, respectively; *α*_1_ is the influence coefficient of the stress on the concrete; *f*_y_*’*, *f*_a_*’*, and *f*_c_ stand for the compressive strength of the steel, reinforcement, and concrete, respectively; *A*_af_ and *A*_af_*’* indicate the flange area of the section steel under tension and compression, respectively; *N*_aw_ and *M*_aw_ are the axial force on and the bending moment of the steel web, respectively; *a* denotes the distance between the resultant point of the rebar and the tensile steel flange to the tensile edge of the column section; *h* is the section height; *e*_0_ indicates the initial eccentricity; and *e*_a_ is the additional eccentricity and is defined as *e*_a_ = max{20 mm, h/30}.

### 5.2. Comparing Calculations of Various Codes with Test Data

Table 7 compares the accuracy of the calculated bearing capacity of different codes, and the calculated results are obtained by EXCEL. Both codes JGJ138-2016 and EN1994-1-1-2004 [18,20] can conservatively predict the bearing capacity of SRHC columns: The calculation results of code JGJ138-2016 are closer to the test data and differ from them on average by less than 10%. Nevertheless, code JGJ138-2016 is unsafe when calculating the load capacity of SRHC columns containing high-strength steel. The calculations of code EN1994-1-1-2004 are more conservative than those of code JGJ138-2016, and the difference between code EN1994-1-1-2004 and the test results is more than 10%, which agrees with the findings of Lai et al. [34]. The calculation results of code AISC360-16 are much lower than the test data, and the difference between its calculations for eccentric columns and the test results is over 30%; thus, code AISC360-16 cannot be directly employed to calculate the bearing capacity of SRHC columns.

In order to explore the accuracy of the calculation results under different parameters, we compare the calculated and experimental results under various parameters in Figure 17. According to Figure 17a, within a steel content range of 3.63–6.20%, the variation trend of the standard calculation results is consistent with that of the test results, indicating no clear relationship between the steel content and the calculation accuracy. Similarly to the steel content, the calculation accuracy of each specification is basically immutable after the stirrup strength, the slenderness ratio, and the eccentricity change (see Figure 17b–d). However, after the stirrup spacing and the steel strength change, the calculation accuracy varies remarkably: As observed in Figure 17e, when the stirrup spacing decreases, the bearing capacity of the columns increases significantly, but the calculations of the specifications remain unchanged. This indicates that the influence of the stirrup constraint on the columns is neglected in the specifications, which makes the calculations exceedingly conservative when the stirrup spacing is small. As shown in Figure 17d, when the steel strength increases beyond 600 MPa, the growth rate of the bearing capacity of the columns tends to decrease markedly, but the calculation results of the specifications still correlate positively with the steel strength. This is because the yield strain of high-strength steel is much higher than that of concrete, and the cover concrete fails before the steel yields. However, the specifications consider that steel and concrete function together completely, which leads to unsafe calculations when the steel strength is high.

### 5.3. Correction Based on Stirrup Constraint and Steel Strength

From the above analysis, it can be inferred that ignoring the constraining effect of the stirrups and overestimating the effect of steel are the main reasons for the error in the calculations of the specifications. In view of the above problems, this paper puts forward some suggestions for revising the formulas.

The confinement effect of the stirrups can effectively improve the bearing capacity and strain of concrete. According to Zhao [47], the concrete in SRC columns can be divided into three parts depending on the constraints, as depicted in Figure 18. However, according to the latest research [48], the constraining effect of H-shaped steel on concrete is slight, which can be ignored in calculations. Therefore, only the restraining effect of the stirrups is considered in the theoretical analysis.

Due to the arching effect of the stirrup restraint, the lateral restraining force cannot be fully exerted in the weakly restrained area of the concrete. Therefore, Mander [46] equalizes the effect of stirrups on the core concrete confinement zone to the entire concrete area within the stirrups and introduced the effective constraint coefficient (*k*_e_) to calculate the effective constraining force (*f*_l_’) of the stirrups as follows:(16)$${f^{\prime }}_{l}=\frac{1}{2}k_{e}\rho_{s}f_{\mathrm{yh}}$$
(17)$$k_{e}=\frac{\left(1-\sum \frac{{\left(w^{\prime }\right)}^{2}}{6b_{c}d_{c}}\right)\left(1-\frac{s^{\prime }}{2b_{c}}\right)\left(1-\frac{s^{\prime }}{2d_{c}}\right)}{1-\rho_{\mathrm{cc}}}$$
where *W* is the spacing of the longitudinal bars; *s* indicates the spacing of the stirrups; *b*_c_ and *d*_c_ represent the length and width of the stirrups, respectively; and *ρ*_cc_ stands for the ratio of the longitudinal reinforcement area to the constraint area.

Mander [46] proposed the improvement coefficient (*k*) of concrete in the confined area to calculate the strength of the confined concrete as follows:(18)$$f_{\mathrm{cc}}=kf_{c0}$$
(19)$$k=-1.254+2.254\sqrt{1+7.94\frac{{f^{\prime }}_{l}}{f_{c0}}}-2\frac{{f^{\prime }}_{l}}{f_{c0}}$$
where *f*_cc_ is the strength of the concrete under equivalent action and *f*_c0_ denotes the strength of the unreinforced concrete.

Since the concrete cover is not constrained, this paper proposes the constraint correction coefficient (*k*_ca_) to calculate the equivalent concrete strength of the whole section:(20)$$f_{\mathrm{ca}}=k_{\mathrm{ca}}f_{c0}=\frac{A_{E}}{A_{C}}kf_{c0}$$
where *A*_C_ is the section area of the specimen and *A*_E_ indicates the concrete area of the stirrup-constrained zone.

Since high-strength steel cannot reach the maximum strength simultaneous with concrete, the decrease in the bearing capacity of the columns caused by the early crushing of concrete must be considered. Figure 19 shows the damage degree to concrete when the steel with different strengths yields. It can be seen that increasing the steel strength raises the degree of the compressive damage to the concrete protective layer remarkably. Therefore, according to the difference between the strains of steel and concrete, this paper introduces the concrete damage reduction coefficient (*γ*) to evaluate the decrease in the bearing capacity of the columns caused by the concrete collapse when steel yields:(21)$$N_{d}=\gamma f_{c}(A_{C}-A_{E})$$
(22)$$\gamma =\frac{\varepsilon_{\Delta }}{\varepsilon_{cu}}\;0\leq \;\gamma \leq 1$$
(23)$$\varepsilon_{\Delta }=\varepsilon_{a}-\varepsilon_{cu}$$
where *N*_d_ represents the decline in the bearing capacity of the column caused by concrete collapse; *A*_C_ indicates the section area of the specimen; *A*_E_ is the concrete area of the stirrup constraint zone; *ε*_a_ stands for the yield strain of steel; and *ε*_cu_ denotes the ultimate compressive strain of concrete.

According to the abovementioned inferences, the strength of the equivalent constrained concrete replaces the concrete strength, and the decrease in the bearing capacity of the column caused by the concrete collapse when the steel yields is considered. Since the concrete of the column under eccentric compression only damages on the compressed side, the reduction in the bearing capacity of the concrete is taken as 1/4*N*_d_. The formulas for calculating the bearing capacity of the column after correction based on code JGJ138-2016 [20] are defined as:(24)$$N\leq 0.9\varphi (k_{\mathrm{ca}}f_{c}A_{c}+{f^{\prime }}_{y}{A^{\prime }}_{s}+{f^{\prime }}_{a}{A^{\prime }}_{a})-N_{d}$$
(25){Nu≤α1kcafcbx+f′yA′s+f′aA′af−σsAs−σaAaf+Naw−14NdNue≤α1kcafcbx(h0−x2)+f′yA′s(h0−a′s)+f′aA′af(h0−a′a)+Maw
(26)$$N_{d}=\gamma f_{c}(A_{C}-A_{E})$$
(27)$$k_{\mathrm{ca}}=\frac{A_{E}}{A_{C}}k$$

Figure 20 compares the accuracy of the revised formula with that of code JGJ138-2016 [20]. The difference between the test results and the calculations of code JGJ138-2016 is about 10%, while the calculations of the revised formula differ from the test data by only about 5%. Thus, the revised formula can accurately estimate the bearing capacity of high-strength concrete columns after considering the stirrup restraint and the decline in the bearing capacity caused by the concrete collapse.

## 6. Conclusions

This work conducted compression tests and Abaqus finite element analysis of steel-reinforced high-strength concrete columns to analyze their failure mode and mechanical properties. In addition, different specifications were compared in terms of their accuracy in calculating the bearing capacity of SRHC columns, and suggestions for improving the accuracy of the calculation methods were proposed. The following conclusions could be drawn from the above findings:All the SRHC columns showed no apparent signs before failure. The analysis of the failure mode of the concrete revealed that the penetration of cracks into aggregate during failure was the main reason for the poor ductility of the SRHC columns.The stiffness of all the SRHC columns remained unchanged before failure, and after reaching the ultimate bearing capacity, the *N–Δ* curves declined dramatically. The constraining effect of the steel on the concrete could also reduce the descending slope of the curve.Increasing the steel content could improve the ductility of the SRHC columns, but the influence was less than 20%. Furthermore, the ductility of the SRHC columns did not correlate positively with their eccentricity. When the eccentricity rises from 0 to 40 and 120 mm, their ductility coefficient enhances by −19% and 28%. The effect on the ductility coefficient by the slenderness ratio is less than 2%.The Abaqus model considering the confinement effect on the concrete could precisely simulate the mechanical properties of the SRHC columns. The finite element analysis demonstrated that the ductility of the SRHC columns markedly improved by reducing the stirrup spacing. The enhance range on the ductility coefficient by the stirrup spacing can exceed than 200%. In addition, increasing the steel strength could enhance the bearing capacity and ductility of the SRHC columns simultaneously.Codes JGJ138-2016, EN1994-1-1-2004, and AISC360-16 underestimated the bearing capacity of the SRHC columns. The effects of the stirrup restraint and the decrease in the bearing capacity caused by the concrete collapse were the main factors affecting the accuracy of the code calculations. The correcting suggestions proposed herein could effectively enhance the accuracy of the specification methods for calculating the bearing capacity of steel-reinforced high-strength concrete columns.

## 7. Recommendations for Future Research

In the current research, compression tests and Abaqus finite element analysis are carried out on the SRHC columns. The measures on enhancing ductility and revisions on the calculation methods are proposed. However, further research of SRHC columns can provide more in-depth investigation on the following:In this paper, measures to enhance the ductility of SRHC columns are proposed, but the solution to premature cracking of concrete cover has not been studied. While the addition of fibers in concrete can effectively hinder concrete cracking and improve the ductility of concrete. As a result, further research can be added in SRHC columns containing fiber.Limited by experimental conditions, the current research on the ductility of SRHC columns under eccentric load is insufficient. More parameters, such as steel form, steel strength, and cross section form can be investigated by experiment and numerical model on the SRHC columns under eccentric load.

## Figures and Tables

**Figure 1 materials-15-06863-f001:**
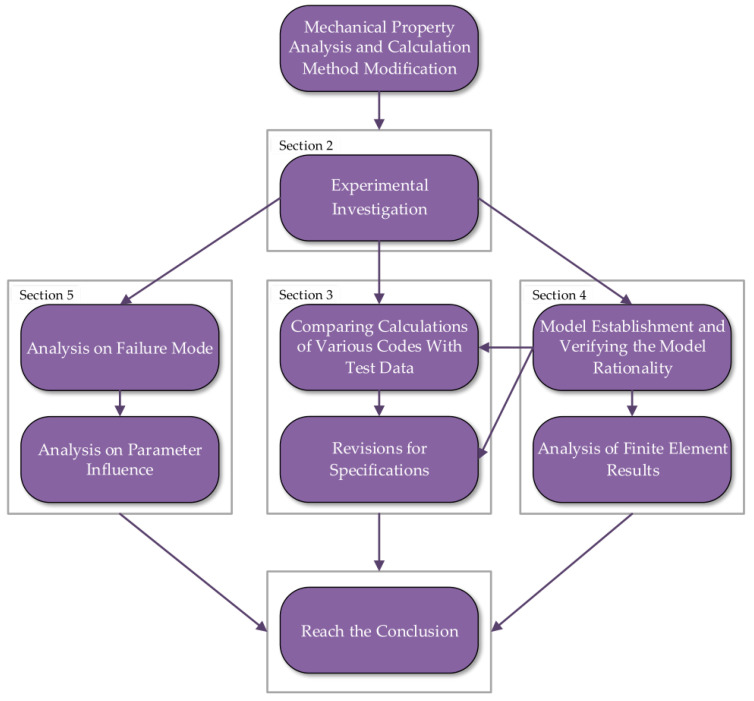
The main research methods and processes.

**Figure 2 materials-15-06863-f002:**
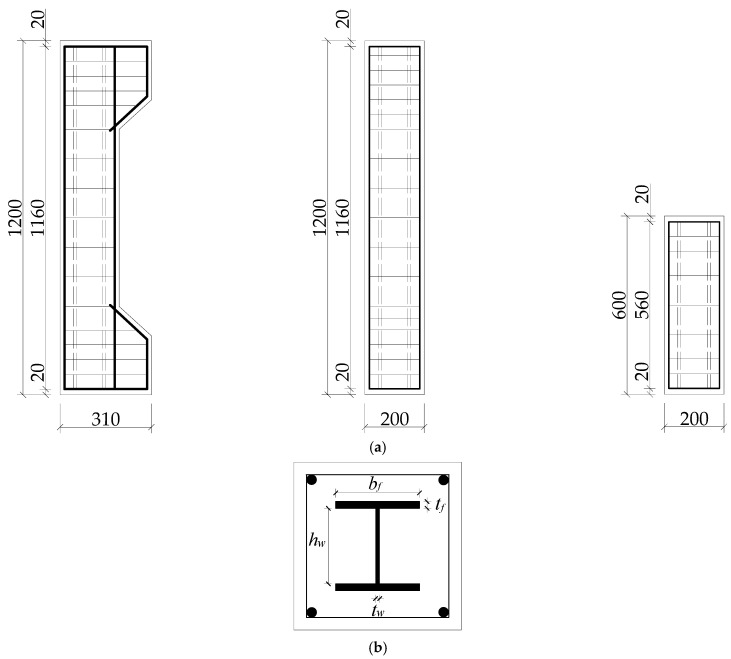
The (**a**) detailed dimensions and (**b**) cross section of the columns.

**Figure 3 materials-15-06863-f003:**
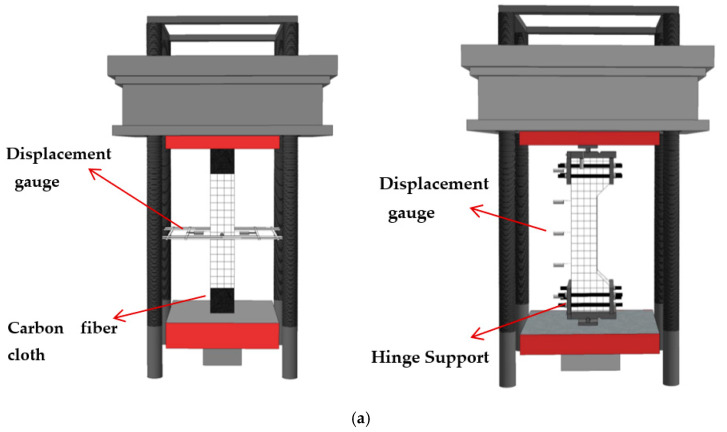
(**a**) A schematic and (**b**) photo of the loading device.

**Figure 4 materials-15-06863-f004:**
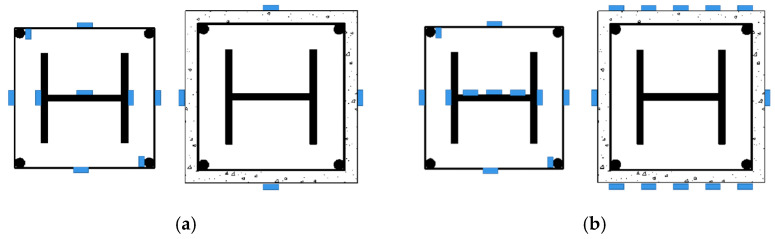
The arrangement of the strain gauges under (**a**) axial and (**b**) eccentric compression.

**Figure 5 materials-15-06863-f005:**
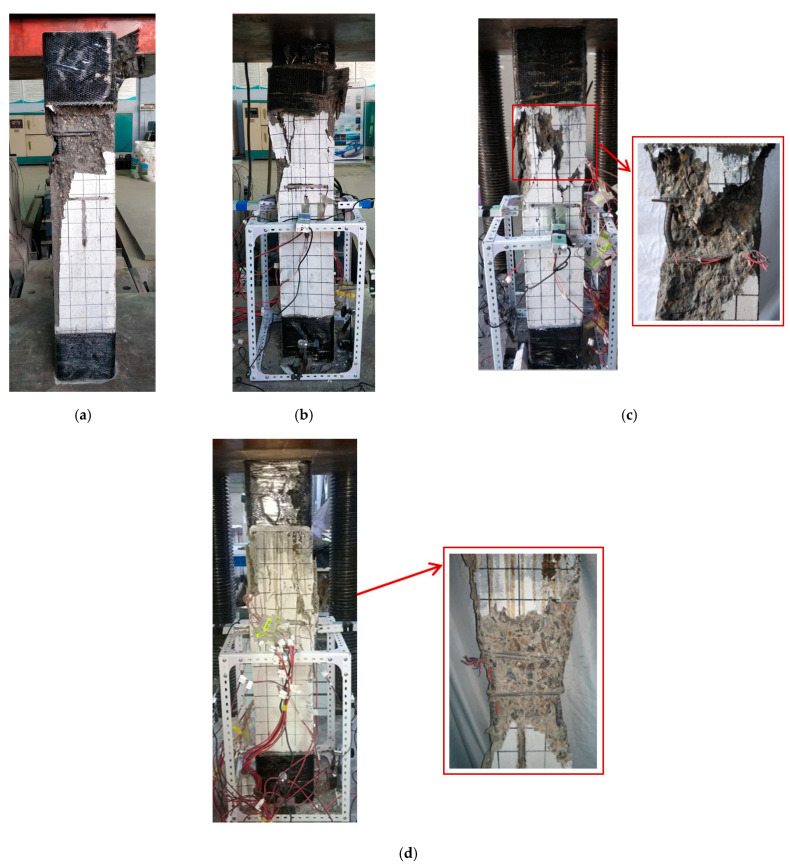
The failure mode of the columns under axial compression: (**a**) specimen A1; (**b**) specimen A2; (**c**) specimen A3; and (**d**) specimen A4 (C50).

**Figure 6 materials-15-06863-f006:**
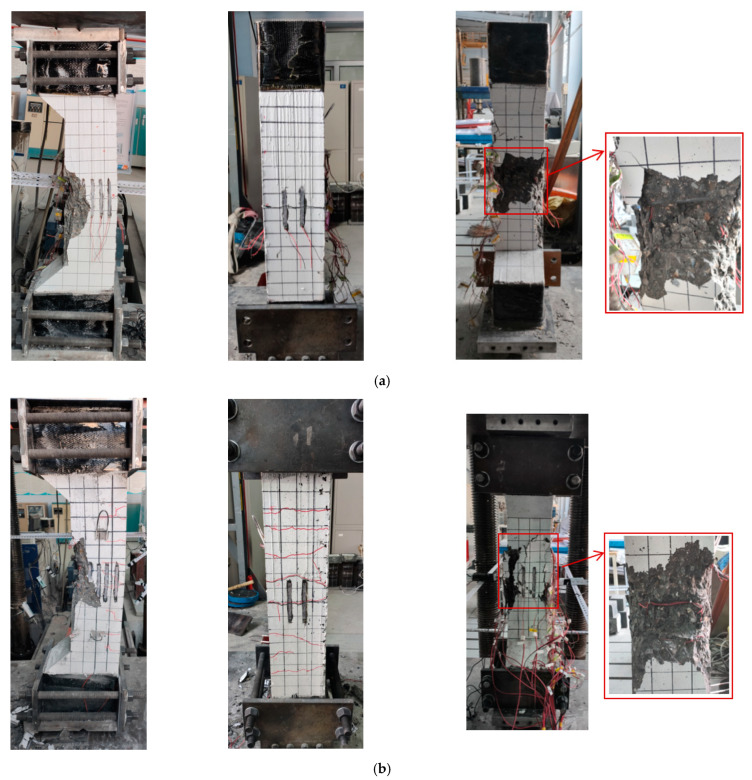
The failure modes of the columns with various eccentric distances: (**a**) specimen E1 with an eccentric distance of 40 mm; (**b**) specimen E2 with an eccentric distance of 120 mm.

**Figure 7 materials-15-06863-f007:**
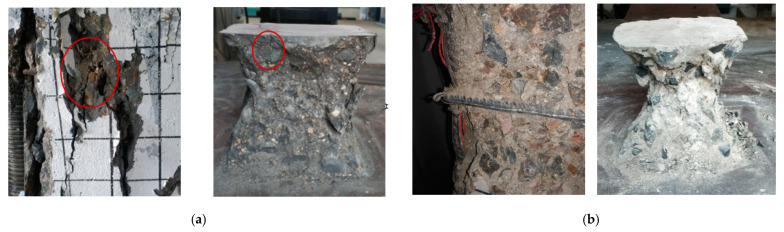
The aggregate failure mode of the specimens and blocks with different concrete strengths: (**a**) C100; (**b**) C50.

**Figure 8 materials-15-06863-f008:**
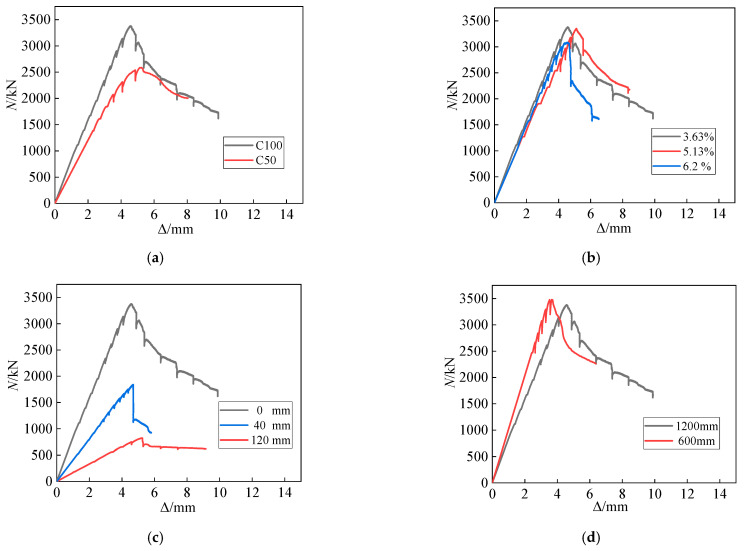
The load–displacement curves of the SRHC columns at various (**a**) concrete strengths, (**b**) steel contents, (**c**) eccentric distances, and (**d**) slenderness ratios.

**Figure 9 materials-15-06863-f009:**
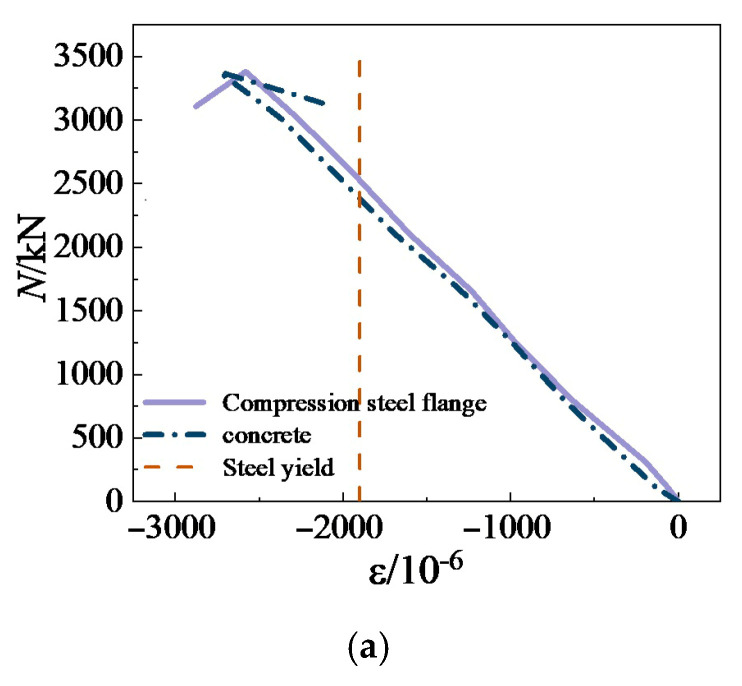
The strain curves of the steel and concrete of the typical specimens: (**a**) specimen A3; (**b**) specimen E1; (**c**) specimen E2.

**Figure 10 materials-15-06863-f010:**
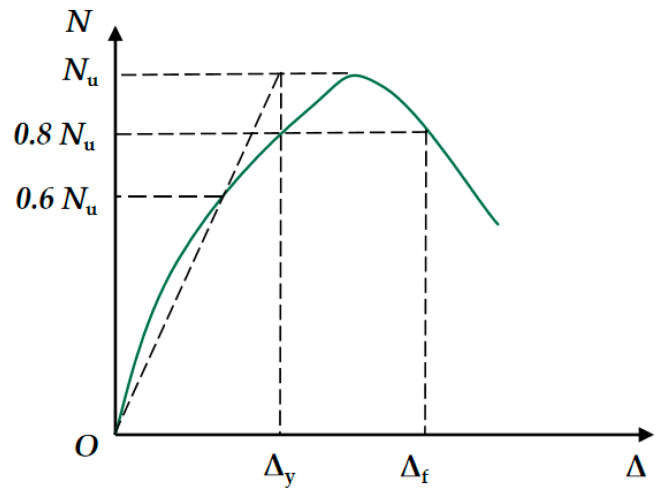
The meaning of the displacement ductility coefficient.

**Figure 11 materials-15-06863-f011:**
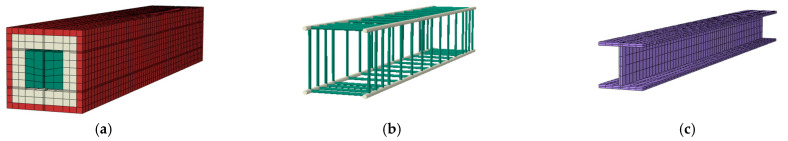
The establishment of the finite element model: (**a**) the concrete part; (**b**) the rebar part; (**c**) the section steel.

**Figure 12 materials-15-06863-f012:**
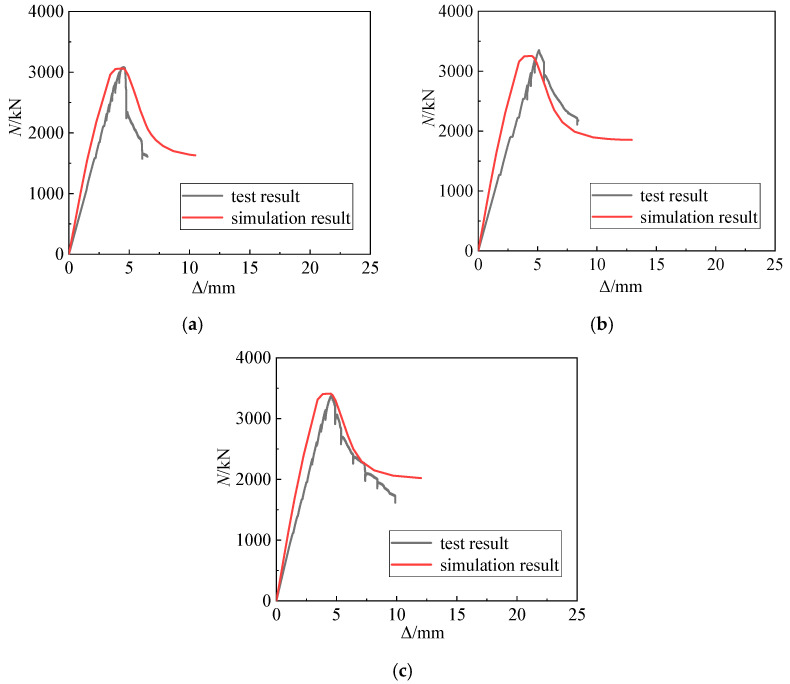
Comparing the simulation curves with the test curves: (**a**) specimen A1; (**b**) specimen A2; (**c**) specimen A3.

**Figure 13 materials-15-06863-f013:**
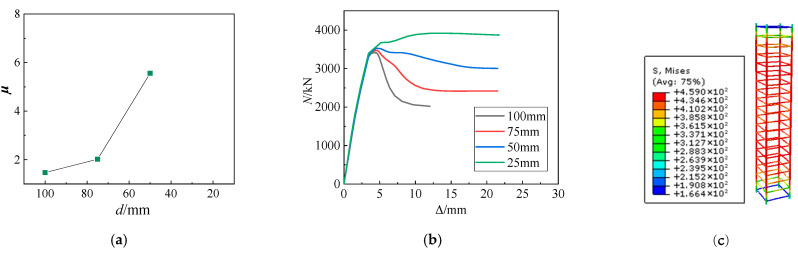
The simulation results at different stirrup spacings: (**a**) the ductility coefficient at different stirrup spacings; (**b**) the *N*–Δ curves at various stirrup spacings; (**c**) the rebar stress nephogram of specimen S3.

**Figure 14 materials-15-06863-f014:**
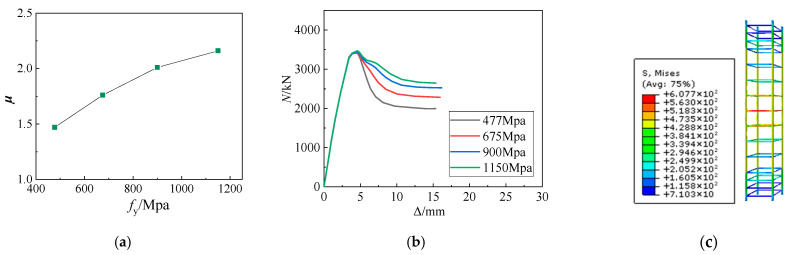
The simulation results at different stirrup strengths: (**a**) the ductility coefficient at different stirrup strengths; (**b**) the *N*–Δ curves at different stirrup strengths; (**c**) the rebar stress nephogram of specimen S7.

**Figure 15 materials-15-06863-f015:**
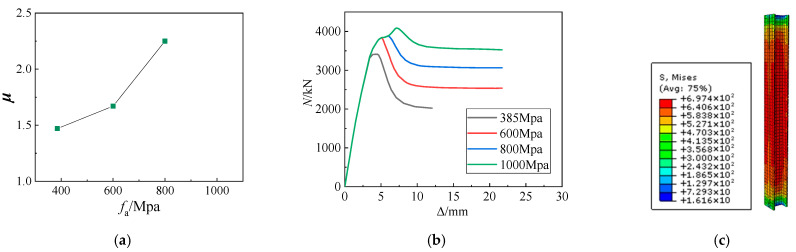
The simulation results at different steel strengths: (**a**) the ductility coefficient at various steel strengths; (**b**) the *N*–Δ curves at different steel strengths; (**c**) the steel stress nephogram of specimen S10.

**Figure 16 materials-15-06863-f016:**
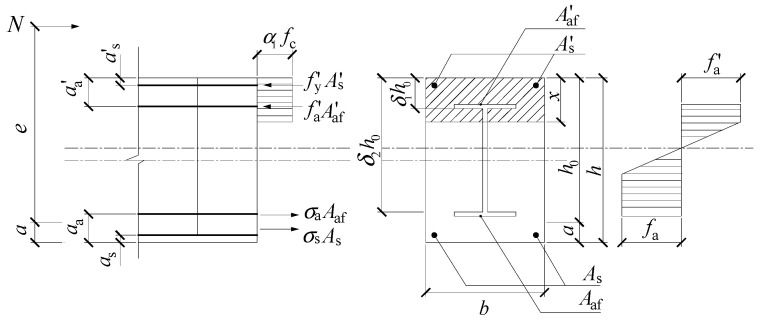
The stress distribution based on code JGJ138-2016.

**Figure 17 materials-15-06863-f017:**
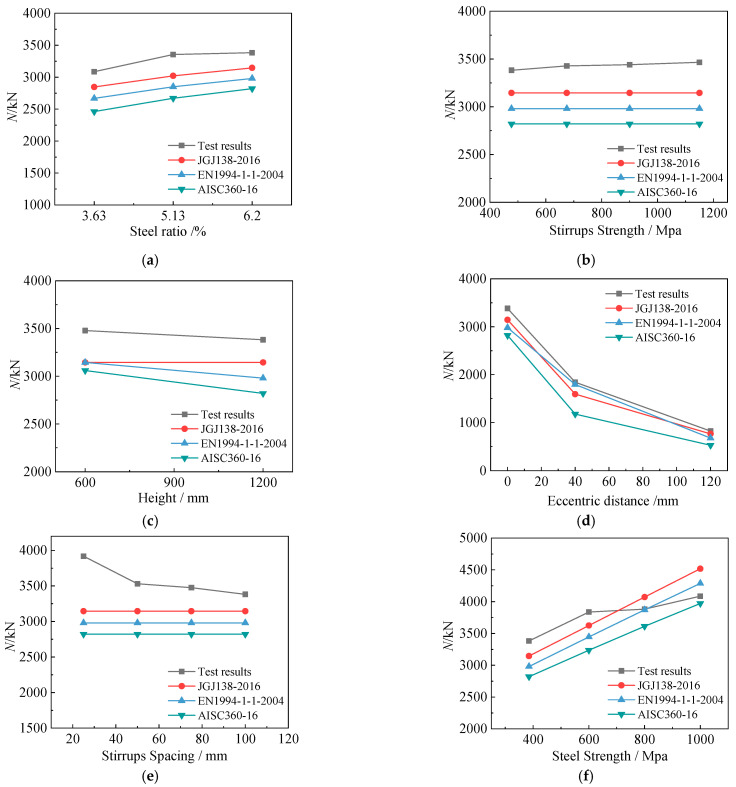
The effects of various parameters on the calculation accuracy: (**a**) the steel ratio; (**b**) the stirrup strength; (**c**) the height of the column; (**d**) the eccentric distance; (**e**) the stirrup spacing; (**f**) the steel strength.

**Figure 18 materials-15-06863-f018:**
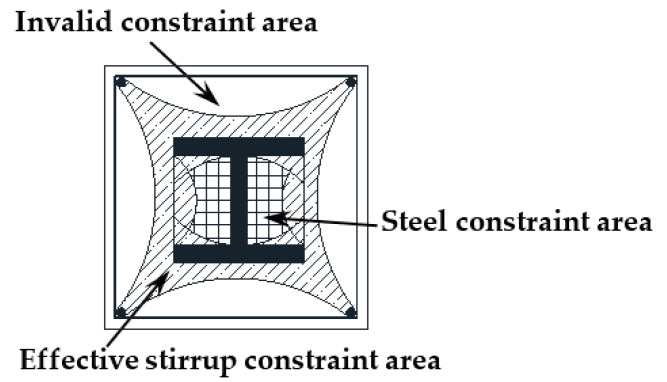
The division of concrete in SRC columns.

**Figure 19 materials-15-06863-f019:**
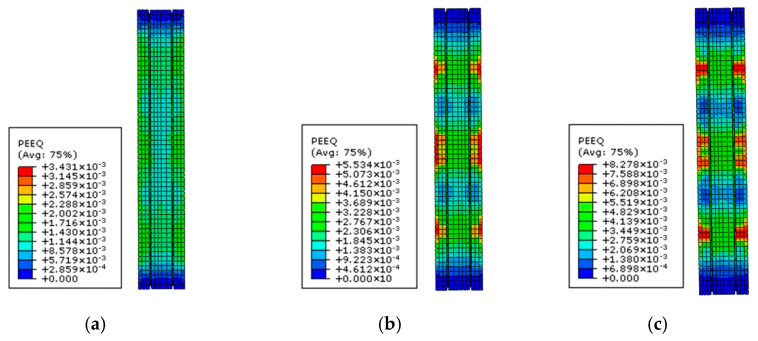
The damage degree to the concrete when the steels with strengths of (**a**) 600 MPa, (**b**) 800 MPa, and (**c**) 1000 MPa yield.

**Figure 20 materials-15-06863-f020:**
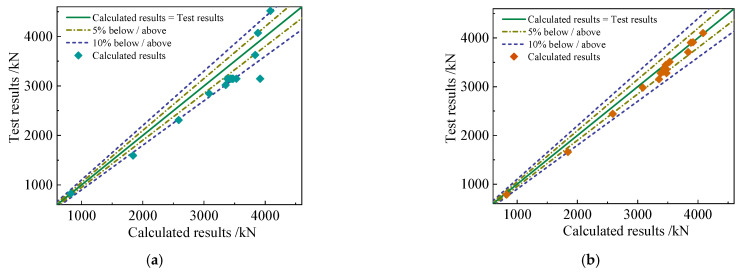
Comparing the calculation results of (**a**) code JGJ138-2016 and (**b**) the revised formula with the test data.

**Table 1 materials-15-06863-t001:** The key parameters of the columns.

Specimen Designation	Concrete Grade	Steel Grade	Dimensions of Steel *h*_w_ × *b*_f_ × *t*_w_ × *t*_f_	Steel Content, *ρ*	Eccentric Distance, *e* (mm)	Height (mm)	Stirrups Spacing, *d* (mm)
AT	C100	Q355	90 × 110 × 8 × 8	6.2%	0	600	100
A1	C100	Q355	90 × 100 × 5 × 5	3.63%	0	1200	100
A2	C100	Q355	90 × 100 × 5 × 8	5.13%	0	1200	100
A3	C100	Q355	90 × 110 × 8 × 8	6.2%	0	1200	100
A4	C50	Q355	90 × 110 × 8 × 8	6.2%	0	1200	100
E1	C100	Q355	90 × 110 × 8 × 8	6.2%	40	1200	100
E2	C100	Q355	90 × 110 × 8 × 8	6.2%	120	1200	100

**Table 2 materials-15-06863-t002:** The C100 concrete mixture design.

Content of Each Component (kg/m^3^)							Water-To-Binder Ratio
Cement	Fly Ash	Silicon Powder	Sand	Pebble	Water	Water Reducer	
455	150	65	638	1015	150	9.75	0.23

**Table 3 materials-15-06863-t003:** The mechanical properties of the steel.

Grade	Specification	Yield Strength, *f*_y_ (MPa)	Ultimate Strength, *f*_u_ (MPa)	Elastic Modulus, *E*_s_ (GPa)
HRB400	8 mm	477	561	214
	12 mm	452	550	208
Q355	5 mm	401	507	206
	8 mm	385	510	198

**Table 4 materials-15-06863-t004:** The test results.

Specimen	*N*_u_ (kN)	Δ_y_ (mm)	Δ_f_ (mm)	*μ*
AT	3478	3.41	4.39	1.28
A1	3085	4.26	4.76	1.07
A2	3354	5.03	6.22	1.23
A3	3382	4.37	5.69	1.30
A4	2586	4.25	7.42	1.75
E1	1841	4.43	4.69	1.06
E2	823	4.71	6.44	1.36

**Table 5 materials-15-06863-t005:** Comparing the simulation data with the test results.

Specimen	Test Results	Simulation Results	Error Rate		
	*N*_u_ (kN)	*N*_s_ (kN)	*N*_s_/*N*_u_–1	Average	Standard Deviation
A1	3085	3062	−0.75%	−0.93%	1.56%
A2	3354	3256	−2.92%		
A3	3382	3412	0.89%		

**Table 6 materials-15-06863-t006:** The simulation results of the finite element expansion parameters.

Specimen	Concrete Grade	Steel Strength, *f*_a_ (MPa)	Stirrups Spacing, *d* (mm)	Stirrups Strength, *f*_y_ (MPa)	Simulation Results, *N*_s_ (kN)	Ductility Coefficient, *μ*
S1	C100	385	100	477	3412	1.47
S2	C100	385	75	477	3476	2.02
S3	C100	385	50	477	3530	5.56
S4	C100	385	25	477	3918	-
S5	C100	385	100	675	3427	1.76
S6	C100	385	100	900	3440	2.01
S7	C100	385	100	1150	3465	2.16
S8	C100	600	100	477	3837	1.67
S9	C100	800	100	477	3882	2.21
S10	C100	1000	100	477	4086	-

**Table 7 materials-15-06863-t007:** Comparing the calculation results of various codes with the test data.

Specimen	Test Results	JGJ138-2016			EN1994-1-1-2004			AISC360-16		
	*N*_u_ (kN)	*N*_J_ (kN)	Error Rate		*N*_E_ (kN)	Error Rate		*N*_A_ (kN)	Error Rate	
			*N*_J_/*N*_u_–1	Average		*N*_E_/*N*_u_–1	Average		*N*_A_/*N*_u_–1	Average
AT	3478	3145	−9.6%	−7.6%	3145	−9.6%	−11.6%	3060	−12.0%	−16.6%
A1	3085	2846	−7.7%		2668	−13.5%		2461	−20.2%	
A2	3354	3020	−10.0%		2850	−15.0%		2670	−20.4%	
A3	3382	3145	−7.0%		2980	−11.9%		2820	−16.6%	
A4	2586	2311	−10.6%		2265	−12.4%		2162	−16.4%	
S2	3476	3145	−9.5%		2980	−14.3%		2820	−18.9%	
S3	3530	3145	−10.9%		2980	−15.6%		2820	−20.1%	
S4	3918	3145	−19.7%		2980	−23.9%		2820	−28.0%	
S5	3427	3145	−8.2%		2980	−13.0%		2820	−17.7%	
S6	3440	3145	−8.6%		2980	−13.4%		2820	−18.0%	
S7	3465	3145	−9.2%		2980	−14.0%		2820	−18.6%	
S8	3837	3625	−5.5%		3446	−10.2%		3237	−15.6%	
S9	3882	4071	4.9%		3871	−0.3%		3612	−7.0%	
S10	4086	4518	10.6%		4288	4.9%		3971	−2.8%	
E1	1841	1594	−13.4%	−10.2%	1794	−2.6%	−10.1%	1177	−36.1%	−35.9%
E2	823	765	−7.0%		677	−17.7%		528	−35.8%	

## Data Availability

The data presented in this study are available on request from the corresponding author.

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
