# Peer review of "Mechanical Property Analysis and Calculation Method Modification of Steel-Reinforced High-Strength Concrete Columns"

_materials, 2022, doi:10.3390/ma15196863_

Round 1
Reviewer 1 Report
Manuscript ID: materials-1923560
Title: Mechanical Property Analysis and Calculation Method Modification of Steel-Reinforced High-Strength Concrete Columns
Journal: Materials
Comments to authors:
The present study aims to test six steel-reinforced high-strength concrete (SRHC) columns and one steel-reinforced concrete (SRC) column to examine the effects of the steel content, eccentric distance, and slenderness ratio on the ductility, bearing capacity, and failure mode of SRHC columns. Further, ABAQUS finite element models are established to analyze the influences of the stirrup spacing, stirrup strength, and steel strength on the ductility of steel-reinforced high- strength concrete columns. This is well-written manuscript with limited novelty. However, to improve the quality of the manuscript, please address the following comments:
1) Some information about the experimental methodology and problem statement should be highlighted before the results section in the Abstract.
2) No need to explain all the results in Abstract.
3) Please add some numeric results in the Abstract.
4) The novelty and significance of the present work should be highlighted in the last paragraph of the Introduction section. The provided reasons are not strong enough.
5) The authors are recommended to add latest relevant literature review on such works.
6) Please add references for codes ‘EN1994-1-1-2004 and JGJ138-2016’.
7) What is the need for this work? For which practical applications, the present work is helpful?
8) The literature review should be improved by adding latest references and discussion.
9) Why no control sample having no steel?
10) Is the fabricated column sample is more suitable and applicable than normal steel-reinforced concrete column having steel rebars?
11) Please add standard deviation of concrete strength to section 2.2.
12) How the post-peak damaging behavior of concrete was simulated in ABAQUS?
13) What about the simulations of steel section in finite element modeling? Which model was used?
14) Work methodologies need more discussion.
15) Please compare the experimental results with the proposed finite element model.
16) The authors are recommended to add parametric study section to investigate the effect of various parameters of fabricated columns using proposed FEA model.
17) How the authors can visualize cracking of concrete through PEEQ strains? Why not principal strains?
18) What about the boundary conditions of samples in ABAQUS?
19) Finite element modeling is also recommended along with the theoretical analysis.
20) More discussion on the post-peak behavior of fabricated beams is required (although authors used load-control technique for tests).
21) How can authors use Mander et al. model for confined concrete during theoretical analysis? This model was proposed for fully confined concrete, instead unconfined one?
22) Conclusions should be refined and elaborated. More numerical results should be added.
23) Future recommendations can be added.
Reviewer 2 Report
The proposed work focuses on the Mechanical Property Analysis and Calculation Method Modification of Steel-Reinforced High-Strength Concrete Columns. It is of potential interest to Materials journal readers.
Despite the importance of the subject addressed, this work needs many improvements to be ready for the publication in the Materials journal.
Specific points of improvement :
- Abstract is too long. So, it must be rewritten
- Literature review section must be improved by more previous researches.
- The objective of this research must be more developed.
- How you can explain that the penetration of cracks into aggregate during failure owing to the high strength of the cementitious materials of concrete is the primary reason for the poor ductility of the SRHC columns.
- Quality of figures must be improved.
- All test standards must be indicated in the manuscript.
- The conclusion section is too long. Only the main results must be indicated.
Reviewer 3 Report
Dear Authors,
you present a very nice and well prepared article.
The experimental and numerical analysis of Steel-Reinforced High-Strength Concrete Columns contains all the essential parts.
In my opinion, the outline of the paper is in order, the figures are well chosen and illustrate all the information.
The text of the thesis is well put together, but the language is lacking.
Perhaps you might consider a deeper revision of the language.
Specific comments:
The introductory chapter lacks citations to some standards.
The introduction also does not include more information on appropriate sustainability HPC structure and the like. See e.g:
10.1088/1755-1315/444/1/012021
10.1617/s11527-020-01535-3
I found the description of the columns, the experiment, the material and all the information and I appreciate it very much.
The 5 photos could be bigger as they are very important for the evaluation.
Similarly all other photos with violations.
The graphs show interesting results from the measurements - it is hugely valuable how in depth you describe this.
I haven't found what software you use for evaluation and calculations?
You are using a lot of extraneous techniques, so the benefit and novelty of the paper is not entirely clear.
Round 2
Reviewer 1 Report
The authors have well addressed the comments. It is ready for publication.
Author Response
Dear reviewer, we are very honored to receive your approval of this article. Thank you very much for agreeing to the publication of this article
Reviewer 2 Report
The proposed work focuses on the Mechanical Property Analysis and Calculation Method Modification of Steel-Reinforced High-Strength Concrete Columns. It is of potential interest to Materials journal readers.
Despite the importance of the subject addressed, this work needs many improvements to be ready for the publication in the Materials journal.
I think that the revised version of the submitted paper is well improved by considering the reviewers and editor recommendations and remarks. Indeed, I think that this paper is accepted in this form
Author Response

(The authors gave the same response as above.)

Reviewer 3 Report
I don't think you have sufficiently reflected my comments. Some of them are obviously a matter of differences of opinion, but in general the article is insufficient for publication. Please study all the comments in detail and try to think about them again.
